# Identification of Small-Molecule Bioactive Constituents from the Leaves of *Vaccinium bracteatum* Confirms It as a Potential Functional Food with Health Benefits

**DOI:** 10.3390/foods12010177

**Published:** 2023-01-01

**Authors:** Yin-Yin Wang, Jun-Sheng Zhang, Xin-Xin Wang, Lin-Lin Tian, Yu-Peng Li, Chao Wang, Ren-Fen Ma, Yi-Ke Yin, Jie Bao, Hua Zhang

**Affiliations:** 1School of Chemistry and Chemical Engineering, University of Jinan, Jinan 250022, China; 2School of Biological Science and Technology, University of Jinan, Jinan 250022, China

**Keywords:** *Vaccinium bracteatum*, antioxidant activity, free radical scavenging activity, neuroprotective activity, iridoid, lignan, functional food

## Abstract

The health benefits of *Vaccinium bracteatum* are well recorded in ancient Chinese medical books and were also demonstrated by modern researches. However, the relationship between its beneficial functions and specific chemical constituents has not been fully characterized. This study investigated the bioactive small-molecule constituents in the leaves of *V. bracteatum*, which afforded 32 compounds including ten new ones (**1**–**9**) and ten pairs of enantiomers (**9**–**18**). Their structures with absolute configurations were elucidated by spectroscopic methods, especially nuclear magnetic resonance (NMR) and electronic circular dichroism (ECD) analyses, with **1**–**4** bearing a novel revolving-door shaped scaffold. While half-compounds exhibited decent antioxidant activity by scavenging 2,2-diphenyl-1-picrylhydrazyl (DPPH) radicals, all except **19** and **20** exerted significant capturing activity against diammonium 2,2′-azino-bis(3-ethylbenzothiazoline-6-sulfonate) (ABTS) radicals. In addition, the new iridoids **1**, **5**, **6**, and **7** exerted apparent neuroprotective activity toward PC12 cells, with **1** being comparable to the positive control, and selective compounds also displayed anti-diabetic and anti-inflammatory properties by inhibiting α-glucosidase and NO production, respectively. The current work revealed that the bioactive small-molecule constituents could be closely related to the functional food property of the title species.

## 1. Introduction

The genus *Vaccinium* Linn. represents a large group of shrubs or small arbors with about 450 species distributed worldwide [1]. Some *Vaccinium* plants, such as blueberries (e.g., *V. corymbosum*, *V. angustifolium*) and bilberry (*V. myrtillus*), are well-known berry trees with important economical values [1,2]. *V. bracteatum* Thunb. is a native Chinese species that mainly grows in the subtropical and tropical regions, especially around the Yangtze river basin, while it is also found in Korea, Japan, Indonesia, and countries on the Indo-China Peninsula [3]. For over a thousand years, since the Northern and Southern Dynasties of China, the leaves of *V. bracteatum* (VBL) have been used as both functional food and natural medicine to nourish the body and enhance longevity [4]. Till now, local people of the Yangtze river basin also consume VBL as food preservative and food colorant, and particularly they have been keeping the custom of eating ‘Wu mi’ (or ‘Wu fan’) every spring, which, as a popular cereal product, is a traditional food made from rice and VBL juice. In the last several years, Wang and coworkers have systematically studied the chemistry involved in the preparation of ‘Wu mi’ from several aspects [5,6,7], which has significantly accelerated the research on *V. bracteatum*. Nevertheless, the small-molecule bioactive constituents corresponding to its traditional applications as functional food still remain to be extensively explored.

Recently, Fan et al. presented a comprehensive review on *V. bracteatum*, with a focus on its health-promoting properties such as hypoglycemic, anti-inflammatory, and antioxidant activities of the extracts or specific constituents [2]. Regarding the bioactive compounds from this species, it was demonstrated that the leaves contain much more metabolites than the fruits, and the major types of compounds include flavones and their glycosides, anthocyanins, terpenoids, iridoid glycosides, phenylpropanoids, and lignans [2]. As a continuation of our recent years’ research efforts on the bioactive ingredients from traditional species with dietary values [8,9], the present work aimed to further explore the bioactive small-molecule compounds from VBL and to help characterize the correlation between its metabolites and traditional health-promoting applications. This study specially concentrated on the isolation and structure characterization of both new and known constituents (Figure 1) from VBL ethanol extract, as well as a comprehensive biological evaluation including antioxidant, neuroprotective, anti-inflammatory, antimicrobial, and α-glucosidase and acetylcholinesterase inhibitory assessments on all the isolates.

## 2. Materials and Methods

### 2.1. Instrumentation and Reagents

See Appendix A. 

### 2.2. Plant Materials

The leaves of *Vaccinium bracteatum* Thunb. were collected at the end of July 2020 in Wuxi, Jiangsu province, China, and the plant materials were identified by Prof. Guo-hua Ye from Shandong College of Traditional Chinese Medicine. The voucher specimen was deposited at the authors’ affiliation (Accession number: npmc-058).

### 2.3. Extraction and Isolation

The dried and powdered leaves of *V. bracteatum* (10.0 kg) were percolated with 20 L 95% EtOH at room temperature for three times, and the solvents were removed *in vacuo* to afford a crude extract (1.3 kg). The crude extract was dissolved in 2.0 L distilled water and partitioned with EtOAc (2.0 L × 3). The EtOAc partition (640 g) was subjected to column chromatography (CC) on D101-macroporous absorption resin (30%, 50%, 80%, and 95% EtOH-H_2_O) to furnish four fractions (A–D). Fraction A (100 g) was divided into five subfractions (A1–A5) by a silica gel column (petroleum ether (PE)-EtOAc, 15:1 to 1:5), and fraction A1 (180 mg) was further fractionated by silica gel CC (CH_2_Cl_2_-MeOH, 100:1 to 10:1) to afford eight fractions (A1-1–A1-8). Fraction A1-2 was purified by HPLC (35% MeOH-H_2_O) to obtain compound **20** (49.6 mg, t_R_ = 12.0 min). Fraction A1-3 was first separated on silica gel CC (PE-acetone, 10:1 to 1:2) and then purified by HPLC (30% MeOH-H_2_O) to yield **17** (2.6 mg, t_R_ = 11.0 min). Fraction A2 (22 g) was separated by the MCI CC (MeOH-H_2_O, 20% to 100%) to furnish six subfractions (A2-1–A2-6) based on TLC analyses. Subfraction A2-1 was first separated by Sephadex LH-20 CC (in CH_2_Cl_2_-MeOH, 1:5) to give three subfractions (A2-1-a–A2-1-c), and A2-1-c was then fractionated by HPLC (40% MeOH-H_2_O) to yield compounds **8** (18.2 mg, t_R_ = 20.5 min), **13** (6.6 mg, t_R_ = 22.5 min), **5** (56.1 mg, t_R_ = 24.0 min), and **14** (8.1 mg, t_R_ = 29.5 min), along with a mixture (21.9 mg, t_R_ = 26.0 min) that was further separated by HPLC (45% CH_3_CN-H_2_O) to afford **15** (2.6 mg, t_R_ = 16.0 min) and **6** (12.1 mg, t_R_ = 21.0 min). The A2-4 subfraction was first separated by silica gel CC (CH_2_Cl_2_-MeOH, 80:1 to 15:1) to afford four subfractions (A2-4-a–A2-4-d), the second one of which was further purified by Sephadex LH-20 CC (in MeOH) and HPLC (50% CH_3_CN-H_2_O) to yield compounds **2** (2.9 mg, t_R_ = 5.5 min) and **1** (19.8 mg, t_R_ = 7.0 min). The A2-5 subfraction was separated by Sephadex LH-20 CC (in MeOH) to obtain three subfractions (A2-5-a–A2-5-c), and compounds **4** (3.2 mg, t_R_ = 14.0 min) and **3** (3.2 mg, t_R_ = 15.0 min) were obtained from A2-5-a by HPLC separation (55% CH_3_CN-H_2_O). Fraction A3 (1.3 g) was first separated by Sephadex LH-20 CC (in MeOH) to give three subfractions (A3-1–A3-3), and A3-2 was then fractionated by RP-C_18_ silica gel CC (MeOH-H_2_O, 25% to 95%) to afford four further fractions (A3-2-a–A3-2-d). Fraction A3-2-b was further separated by HPLC (38% MeOH-H_2_O) to yield **16** (2.1 mg, t_R_ = 12.0 min), **12** (6.0 mg, t_R_ = 14.5 min), and **10** (1.6 mg, t_R_ = 16.0 min), while fraction A3-2-d was purified by HPLC (45% MeOH-H_2_O) to afford **19** (12.6 mg, t_R_ = 8.0 min). Fraction A3-3 was processed with the same gradient elution as that of A3-2 to furnish three subfractions (A3-3-a–A3-3-c), and A3-3-a was then fractionated by HPLC (40% MeOH-H_2_O) to afford compounds **18** (3.3 mg, t_R_ = 10.0 min), **22** (4.2 mg, t_R_ = 15.0 min), **9** (4.2 mg, t_R_ = 17.0 min), and **11** (7.0 mg, t_R_ = 19.5 min). Fraction B (70.0 g) was fractionated by silica gel CC (PE-EtOAc, 8:1 to 1:2) to produce four elutions (B1–B4). Elution B1 (4.77 g) was first separated by Sephadex LH-20 CC (in MeOH) to give four subfractions (B1-1–B1-4), and B1-1 was then chromatographed on RP-C_18_ silica gel CC (MeOH-H_2_O, 30% to 95%) to give another four subfractions (B1-1-a–B1-1-d). Compound **7** (1.4 mg, t_R_ = 6.5 min) was then purified from subfraction B1-1-b by HPLC (45% CH_3_CN-H_2_O). Furthermore, compounds **9–18** were identified as ten pairs of enantiomers and eventually separated by HPLC on DAICEL AD-H, OD-H, or IC chiral columns (1.0 mL/min). Compounds **10a** (0.5 mg, t_R_ = 27.0 min)/**10b** (0.5 mg, t_R_ = 29.5 min), **12a** (2.9 mg, t_R_ = 15.5 min)/**12b** (2.4 mg, t_R_ = 17.0 min), and **14a** (3.5 mg, t_R_ = 21.0 min)/**14b** (2.4 mg, t_R_ = 24.0 min) were separated on the AD-H chiral column with the mobile phase of 20% isopropanol in *n*-hexane. Compounds **15a** (0.9 mg, t_R_ = 15.0 min)/**15b** (0.7 mg, t_R_ = 23.0 min), **16a** (0.9 mg, t_R_ = 19.0 min)/**16b** (0.7 mg, t_R_ = 25.5 min), and **17a** (0.8 mg, t_R_ = 15.0 min)/**17b** (0.7 mg, t_R_ = 19.5 min) were separated on the IC chiral column with the mobile phase of 42% isopropanol in *n*-hexane. Compounds **9a** (1.2 mg, t_R_ = 21.0 min)/**9b** (1.1 mg, t_R_ = 23.5 min), **11a** (3.0 mg, t_R_ = 11.0 min)/**11b** (1.6 mg, t_R_ = 12.5 min), and **13a** (0.7 mg, t_R_ = 7.5 min)/**13b** (2.5 mg, t_R_ = 28.5 min) were separated on the IC chiral column with the mobile phase of 27% isopropanol in *n*-hexane. The mobile phase of 35% isopropanol in *n*-hexane with the OD-H chiral column was used to accomplish the isolation of compounds **18a** (0.9 mg, t_R_ = 15.0 min)/**18b** (0.9 mg, t_R_ = 25.0 min), and finally, a subfraction from the mixture with **18a**/**18b** was further separated by the OD-H column (19% isopropanol in *n*-hexane) to afford **21** (0.9 mg, t_R_ = 28.5 min).

#### 2.3.1. Compound **1**

Yellowish gum; ${\left[\alpha \right]}_{D}^{25}$ –29.3 (*c* 0.25, MeOH); UV (MeOH) *λ*_max_ (log *ε*) 313 (4.40), 229 (3.85) nm; ^1^H and ^13^C NMR data (methanol-*d*_4_) (see Appendix A); ESIMS *m/z* 385.1 [M + Na]^+^; HR-ESIMS *m/z* 385.1259 [M + Na]^+^ (calcd for C_19_H_22_O_7_Na^+^, 385.1258).

#### 2.3.2. Compound **2**

Yellowish gum; ${\left[\alpha \right]}_{D}^{25}$ –7.6 (*c* 0.25, MeOH); UV (MeOH) *λ*_max_ (log *ε*) 311 (4.24), 226 (3.74) nm; ^1^H and ^13^C NMR data (methanol-*d*_4_) (see Appendix A); ESIMS *m/z* 747.2 [2M + Na]^+^; HR-ESIMS *m/z* 385.1261 [M + Na]^+^ (calcd for C_19_H_22_O_7_Na^+^, 385.1258).

#### 2.3.3. Compound **3**

Yellowish gum; ${\left[\alpha \right]}_{D}^{25}$ –4.0 (*c* 0.5, MeOH); UV (MeOH) *λ*_max_ (log *ε*) 313 (4.05), 230 (2.94) nm; ^1^H and ^13^C NMR data (methanol-*d*_4_) (see Appendix A); ESIMS *m/z* 370.9 [M + Na]^+^; HR-ESIMS *m/z* 371.1100 [M + Na]^+^ (calcd for C_19_H_22_O_7_Na^+^, 371.1101).

#### 2.3.4. Compound **4**

Yellowish gum; ${\left[\alpha \right]}_{D}^{25}$ –26.1 (*c* 0.2, MeOH); UV (MeOH) *λ*_max_ (log *ε*) 311 (4.22), 226 (3.72) nm; ^1^H and ^13^C NMR data (methanol-*d*_4_) (see Appendix A); ESIMS *m/z* 371.0 [M + Na]^+^; HR-ESIMS *m/z* 371.1095 [M + Na]^+^ (calcd for C_19_H_22_O_7_Na^+^, 371.1101).

#### 2.3.5. Compound **5**

Yellowish gum; ${\left[\alpha \right]}_{D}^{25}$ –82.2 (*c* 0.2, MeOH); UV (MeOH) *λ*_max_ (log *ε*) 313 (4.63), 228 (4.08) nm; ^1^H and ^13^C NMR data (methanol-*d*_4_) (see Appendix A); ESIMS *m/z* 416.9 [M + Na]^+^; HR-ESIMS *m/z* 417.1523 [M + Na]^+^ (calcd for C_20_H_26_O_8_Na^+^, 417.1520).

#### 2.3.6. Compound **6**

White solid; ${\left[\alpha \right]}_{D}^{25}$ –95.4 (*c* 0.125, MeOH); UV (MeOH) *λ*_max_ (log *ε*) 310 (4.44), 226 (3.83) nm; ^1^H and ^13^C NMR data (methanol-*d*_4_) (see Appendix A); ESIMS *m/z* 417.2 [M + Na]^+^; HR-ESIMS *m/z* 417.1525 [M + Na]^+^ (calcd for C_20_H_26_O_8_Na^+^, 417.1520).

#### 2.3.7. Compound **7**

Yellowish gum; ${\left[\alpha \right]}_{D}^{25}$ +15.8 (*c* 0.015, MeOH); UV (MeOH) *λ*_max_ (log *ε*) 312 (4.30), 226 (4.15) nm; ECD (*c* 0.075 mg/mL, MeOH), *λ* (Δ*ε*) 309 (+0.88) nm; ^1^H and ^13^C NMR data (methanol-*d*_4_) (see Appendix A); ^1^H NMR (CDCl_3_, 600 MHz) *δ*_H_ 7.64 (d, *J* = 15.9 Hz, H-7′), 7.40 (d, *J* = 8.5 Hz, H-2′/6′), 6.84 (d, *J* = 8.5 Hz, H-3′/5′), 6.29 (d, *J* = 15.9 Hz, H-8′), 6.00 (dd, *J* = 5.7, 2.4 Hz, H-6), 5.80 (dd, *J* = 5.7, 1.7 Hz, H-7), 5.07 (d, *J* = 5.7 Hz, H-1), 4.85 (dd, *J* = 7.2, 4.5 Hz, H-3), 4.35 (d, *J* = 11.3 Hz, H_a_-10), 4.19 (d, *J* = 11.3 Hz, H_b_-10), 3.50 (s, 3-OC*H*_3_), 3.47 (s, 1-OC*H*_3_), 2.94 (m, H-5), 2.32 (dd, *J* = 8.5, 5.7 Hz, H-9), 2.16 (ddd, *J* = 14.2, 6.2, 4.5 Hz, H_a_-4), 1.49 (ddd, *J* = 14.2, 10.3, 7.2 Hz, H_b_-4); ^13^C NMR (CDCl_3_, 151 MHz); *δ*_C_ 167.7 (C-9′), 158.2 (C-4′), 145.3 (C-7′), 138.7 (C-6), 132.9 (C-7), 130.2 (C-2′/6′), 127.1 (C-1′), 116.1 (C-3′/5′), 115.0 (C-8′), 98.6 (C-3), 97.5 (C-1), 83.9 (C-8), 69.6 (C-10), 55.7 (1-O*C*H_3_), 56.0 (3-O*C*H_3_), 46.6 (C-9), 40.0 (C-5), 33.1 (C-4); ESIMS *m/z* 775.1 [2M + Na]^+^; HR-ESIMS *m/z* 399.1414 [M + Na]^+^ (calcd for C_20_H_24_O_7_Na^+^, 399.1414).

#### 2.3.8. Compound **8**

Yellowish gum; ${\left[\alpha \right]}_{D}^{25}$ +9.5 (*c* 0.2, MeOH); UV (MeOH) *λ*_max_ (log *ε*) 312 (4.61) nm; ^1^H and ^13^C NMR data (methanol-*d*_4_) (see Appendix A); ^1^H NMR (DMSO-*d*_6_, 600 MHz) *δ*_H_ 7.60 (d, *J* = 16.0 Hz, H-7′), 7.55 (d, *J* = 8.6 Hz, H-2′/6′), 6.80 (d, *J* = 8.6 Hz, H-3′/5′), 6.38 (d, *J* = 16.0 Hz, H-8′), 4.79 (d, *J* = 3.5 Hz, H-1), 4.70 (dd, *J* = 5.7, 4.1 Hz, H-3), 3.96 (m, H-6), 4.08 (d, *J* = 10.9 Hz, H_a_-10), 3.97 (d, *J* = 10.8 Hz, H_b_-10), 3.32 (s, 1-OC*H*_3_), 3.33 (s, 3-OC*H*_3_), 2.07 (m, H-9), 2.06 (m, H-5), 1.90 (dd, *J* = 13.1, 6.2 Hz, H_a_-7), 1.59 (dd, *J* = 13.5, 7.8 Hz, H_b_-7), 1.72 (m, H_2_-4); ^13^C NMR (DMSO-*d*_6_, 151 MHz) *δ*_C_ 166.6 (C-9′), 159.9 (C-4′), 144.8 (C-7′), 130.3 (C-2′/6′), 125.1 (C-1′), 115.8 (C-3′/5′), 114.2 (C-8′), 98.6 (C-1), 97.3 (C-3), 77.2 (C-8), 73.4 (C-6), 69.9 (C-10), 55.1 (1-O*C*H_3_), 54.7 (3-O*C*H_3_), 45.4 (C-9), 45.3 (C-7), 40.2 (C-5), 28.8 (C-4); ESIMS *m/z* 416.9 [M + Na]^+^; HR-ESIMS *m/z* 417.1517 [M + Na]^+^ (calcd for C_20_H_26_O_8_Na^+^, 417.1520).

#### 2.3.9. Compound **9**

Colorless gum; ${\left[\alpha \right]}_{D}^{25}$ and ECD data (see Appendix A); UV (MeOH) *λ*_max_ (log *ε*) 278 (4.16) nm; ^1^H NMR (methanol-*d*_4_, 600 MHz) *δ*_H_ 6.81 (d, *J* = 8.2 Hz, H-5′), 6.78 (d, *J* = 1.9 Hz, H-2′), 6.69 (s, H-2/6), 6.66 (dd, *J* = 8.2, 1.9 Hz, H-6′), 4.80 (d, *J* = 6.0 Hz, H-7), 4.30 (m, H-8), 3.85 (dd, *J* = 11.9, 5.8 Hz, H_a_-9), 3.80 (s, 3(5)-OC*H*_3_), 3.78 (s, 3′-OC*H*_3_), 3.77 (m, H_b_-9), 3.54 (t, *J* = 6.5 Hz, H_2_-9′), 2.59 (t, *J* = 7.7 Hz, H_2_-7′), 1.79 (tt, *J* = 7.7, 6.5 Hz, H_2_-8′); ^13^C NMR (methanol-*d*_4_, 151 MHz) *δ*_C_ 151.7 (C-3′), 148.9 (C-3(5)), 147.2 (C-4′), 137.9 (C-1′), 135.9 (C-4), 133.3 (C-1), 121.8 (C-6′), 119.3 (C-5′), 113.9 (C-2′), 105.6 (C-2(6)), 86.4 (C-8), 74.3 (C-7), 62.3 (C-9), 62.2 (C-9′), 56.7 (3(5)-O*C*H_3_), 56.4 (3′-O*C*H_3_), 35.5 (C-8′), 32.7 (C-7′); ESIMS *m/z* 430.9 [M + Na]^+^; HR-ESIMS *m/z* 443.1481 [M + Cl]^–^ (calcd for C_21_H_28_O_8_Cl^–^, 443.1478).

#### 2.3.10. Compound **21**

White solid; ${\left[\alpha \right]}_{D}^{25}$ −65.1 (*c* 0.12, MeOH); UV (MeOH) *λ*_max_ (log *ε*) 283 (4.17) nm; ECD (*c* 0.03, MeOH) *λ* (Δ*ε*) 209 (–15.3) nm. ^1^H NMR (methanol-*d*_4_, 600 MHz) *δ*_H_ 6.63 (d, *J* = 1.9 Hz, H-2′), 6.58 (d, *J* = 8.2 Hz, H-3), 6.56 (d, *J* = 1.9 Hz, H-6), 6.52 (brd, *J* = 8.2 Hz, H-2), 6.47 (d, *J* = 1.9 Hz, H-6′), 3.82 (s, 3′-OC*H*_3_), 3.68 (s, 5-OC*H*_3_), 3.76 (dd, *J* = 10.7, 6.2 Hz, H_a_-9), 3.73 (dd, *J* = 10.7, 6.9 Hz, H_b_-9), 3.51 (t, *J* = 6.5 Hz, H_2_-9′), 3.40 (m, H-8), 2.99 (dd, *J* = 13.7, 5.9 Hz, H_a_-7), 2.87 (dd, *J* = 13.7, 8.9 Hz, H_b_-7), 2.53 (t, *J* = 7.6 Hz, H_2_-7′), 1.74 (tt, *J* = 7.6, 6.5 Hz H_2_-8′); ^13^C NMR (methanol-*d*_4_, 151 MHz) *δ*_C_ 148.7 (C-3′), 148.3 (C-5), 145.2 (C-4), 143.7 (C-4′), 133.72 (C-1), 133.70 (C-1′), 129.4 (C-5′), 122.6 (C-2), 122.0 (C-6′), 115.6 (C-3), 113.8 (C-6), 110.6 (C-2′), 65.8 (C-9), 62.2 (C-9′), 56.5 (3′-O*C*H_3_), 56.2 (5-O*C*H_3_), 45.6 (C-8), 37.5 (C-7), 35.8 (C-8′), 32.8 (C-7′); ESIMS *m/z* 385.2 [M + Na]^+^.

### 2.4. Alkaline Hydrolysis of Compounds ***5*** and ***8***

To a stirred solution of molecule **5** (8 mg) in 2.5 mL methanol was added excess 8% aqueous NaOH (0.2 mL), and the reaction mixture was stirred at 80 °C for 90 min till the finish of hydrolysis monitored by TLC. The reaction solvent was then evaporated, and the residue was purified by a silica gel column (CH_2_Cl_2_-MeOH, 45:1) to give the desired product **5r** (4.0 mg) as a yellowish gum. ^1^H and ^13^C NMR data (CDCl_3_) (see Appendix A). ^1^H NMR (methanol-*d*_4_, 600 MHz) *δ*_H_ 4.95 (d, *J* = 4.9 Hz, H-1), 4.76 (dd, *J* = 8.3, 4.4 Hz, H-3), 4.10 (td, *J* = 6.4, 3.5 Hz, H-6), 3.47 (d, *J* = 11.0 Hz, H_a_-10), 3.46 (d, *J* = 11.0 Hz, H_b_-10), 3.46 (s, 1-OC*H*_3_), 3.45 (s, 3-OC*H*_3_), 2.17 (m, H-5), 2.15 (m, H-9), 2.06 (dd, *J* = 13.9, 6.4 Hz, H_a_-7), 1.98 (m, H_a_-4), 1.77 (dd, *J* = 13.9, 6.4 Hz, H_b_-7), 1.54 (m, H_b_-4); ^13^C NMR (methanol-*d*_4_, 151 MHz) *δ*_C_ 99.5 (C-3), 99.4 (C-1), 81.7 (C-8), 77.1 (C-6), 69.6 (C-10), 56.0 (1-O*C*H_3_), 55.9 (3-O*C*H_3_), 47.4 (C-9), 46.2 (C-7), 44.5 (C-5), 32.5 (C-4).

Compound **8** (10 mg) was processed with the same procedure to afford **8r** (7.7 mg). 1D NMR data (CDCl_3_) (see Appendix A). ^1^H NMR (methanol-*d*_4_, 600 MHz) *δ*_H_ 4.77 (d, *J* = 5.0 Hz, H-1), 4.76 (dd, *J* = 6.8, 3.7 Hz, H-3), 4.11 (q-like, *J* = 7.6 Hz, H-6), 3.455 (s, 1-OC*H*_3_), 3.448 (d, *J* = 11.0 Hz, H_a_-10), 3.445 (s, 3-OC*H*_3_), 3.42 (d, *J* = 11.0 Hz, H_b_-10), 2.19 (dd, *J* = 9.8, 5.0 Hz H-9), 2.16 (m, H-5), 1.98 (dd, *J* = 13.5, 6.7 Hz, H_a_-7), 1.88 (ddd, *J* = 13.6, 5.5, 3.7 Hz, H_a_-4), 1.79 (ddd, *J* = 13.6, 6.9, 6.0, H_b_-4), 1.68 (dd, *J* = 13.5, 8.3, H_b_-7); ^13^C NMR (methanol-*d*_4_, 151 MHz) *δ*_C_ 100.8 (C-1), 99.5 (C-1), 80.4 (C-8), 75.0 (C-6), 69.9 (C-10), 56.02 (1-O*C*H_3_), 55.96 (3-O*C*H_3_), 46.7 (C-9), 45.7 (C-7), 42.6 (C-5), 29.9 (C-4).

### 2.5. Preparation of (S) and (R)-MTPA Esters of ***5*** and ***8***

To a solution of compound **5** (2.0 mg) in pyridine (0.5 mL) was added (*R*)-α-methoxy-α-(trifluoromethyl) phenylacetyl chloride (10 μL) and 4-dimethylaminopyridine (DMAP, 2.0 mg). The reaction mixture was stirred at room temperature for 4.5 h, and the crude product acquired after removal of the solvent was further purified by HPLC to give the (*S*)-MTPA ester **5a** (0.9 mg). The (*R*)-MTPA ester **5b**, as well as the (*S*) and (*R*)-MTPA esters **8a** and **8b** of **8**, was prepared via the same protocol.

### 2.6. ECD Calculation

See Appendix A.

### 2.7. Bioassays

#### 2.7.1. Antioxidant Assay (ABTS and DPPH Radical Scavenging Assays)

The antioxidant effect of the VBL constituents was assessed by besting their free radical scavenging activity with the ABTS (diammonium 2,2′-azino-bis (3-ethylbenzothiazoline-6-sulphonate) and DPPH (1,1-diphenyl-2-picrylhydrazyl) methods, according to the protocols described previously [10], and ascorbic acid was used as the positive control.

#### 2.7.2. NO Production Inhibitory Assay

The nitric oxide (NO) production inhibitory assay was conducted in murine RAW264.7 macrophages as we recorded in an early report [11], with quercetin as the reference compound.

#### 2.7.3. Acetylcholinesterase Inhibitory Assay

The acetylcholinesterase (AChE) inhibitory assay was carried out according to a method reported formerly [12], and tacrine was applied as the control drug.

#### 2.7.4. α-Glucosidase Inhibitory Assay

The α-glucosidase inhibitory effect of all the molecules was tested by a previously described method [13], with acarbose as the reference drug.

#### 2.7.5. Antibacterial Assay

The antibacterial activities of all the compounds against the Gram+ strains *Staphylococcus aureus* ATCC 25923 and *Bacillus subtili*s ATCC 6633, as well as the Gram– strains *Pseudomonas aeruginosa* ATCC 9027 and *Escherichia coli* ATCC 8739, were screened using the liquid growth inhibition method as formerly documented [14]. Ceftriaxone sodium was used as the reference compound for the current assay.

#### 2.7.6. Neuroprotective Assay

The cell cultures were prepared according to the method described by Sergi et al. [15]. Briefly, the protection of compounds **1**–**8** on 6-hydroxydopamine (6-OHDA)-induced PC 12 cell injury was assessed by MTT (3-(4,5-dimethylthiazol-2-yl)-2,5-diphenyltetrazolium bromide) assay, and *N*-acetylcysteine (NAC, 50 μM) was used as the positive control. PC12 cells were cultured in RPMI 1640 medium supplemented with 10% fetal bovine serum (FBS), and they were incubated at 37 

C in a humidified environment containing 5% CO_2_ and then cultured in 96-well plates at a density of 1.5 × 10^5^ cells/well in 100 μL for 24 h. The neuron cells were first pretreated with various concentrations (25, 50, 100 μM) of tested compounds for 12 h before 6-OHDA (150 μM) treatment for further 13 h. Then, 10 μL of MTT was added to each well and incubated for another 4 h, and the crystals were dissolved with DMSO. The optical density at 490 nm was detected by a plate reader. The cell viability was lastly measured to evaluate the compounds’ protective effect, which was expressed as the percentage of the control group.

## 3. Results

### 3.1. Structure Characterization of New Compounds

The molecular formula C_19_H_22_O_7_ for compound **1** was assigned via NMR data and HR-ESIMS analysis at *m*/*z* 385.1259 ([M + Na]^+^, calcd for 385.1258), indicating nine indices of hydrogen deficiency. The NMR data (Appendix A) for **1** exhibited resonances characteristic for a (*E*)-coumaroyloxy fragment [*δ*_C_ 168.7, 161.3, 147.1, 131.3 (2C), 127.1, 116.8 (2C), 114.7; *δ*_H_ 7.66, 6.36 (both d, *J* = 15.9 Hz), 7.47, 6.81 (both d and 2H, *J* = 8.6 Hz)]. In addition, signals for two acetal methines [*δ*_C_ 98.0, 92.1; *δ*_H_ 5.16 (d, *J* = 3.1 Hz), 4.96 (dd, *J* = 3.1, 1.2 Hz)], an oxygenated sp^3^ quaternary carbon (*δ*_C_ 82.2), an sp^3^ oxymethine [*δ*_C_ 77.3; *δ*_H_ 4.05 (dd, *J* = 6.9, 2.0 Hz)], an sp^3^ oxymethylene [*δ*_C_ 66.3; *δ*_H_ 4.34 and 4.14 (both d, *J* = 11.5 Hz)], and a methoxy group [*δ*_C_ 55.3; *δ*_H_ 3.42 (s)], as well as non *O*-bonded aliphatic signals (*δ*_C_ 46.8, 40.4, 39.9, 34.9; *δ*_H_ 2.56, 2.45, 2.31, 2.25, 1.82, 1.57), were further observed. The aforementioned NMR features accounted for six indices of hydrogen deficiency, and the remaining three suggested a tricyclic backbone for **1**. Further inspection of 2D NMR ^1^H–^1^H COSY and HMBC data (Figure 2) established the planar structure of **1** as described below. Analysis of the ^1^H–^1^H COSY correlations afforded, in addition to the coumaroyloxy moiety, two proton-bearing structural units of CH-6/CH_2_-7 and CH-3/CH_2_-4/CH-5/CH-9/CH-1, which were then connected with other fragments to form the bicyclic iridoid core of **1**, based on the HMBC signals from 1-OC*H*_3_ and H-3 to C-1, H_2_-4 to C-5 and C-6, and H_2_-10 to C-7, C-8, and C-9. The (*E*)-coumaroyloxy unit was linked to C-10 by the HMBC correlations from H_2_-10 to C-9′. Finally, considering the remaining undescribed ring, the HMBC signal from H-3 to C-8, and the chemical shifts for C-3 and C-8, an ether bridge between C-3 and C-8 was established. The relative configuration of **1** was determined by 3D conformational analysis and the examination of NOESY data as shown in Figure 2. Owing to severe steric hindrance, the 3D structure of **1** is very rigid and presents an interesting revolving-door shape (door 1: C-9/C-1/O/C-3; door 2: C-9/C-5/C-4/C-3; door 3: C-9/C-8/O/C-3), which leads to the only possibility for the relative configuration assignments at C-3, C-5, C-8, and C-9. Then, the NOESY correlations of H-1/H-10a and H-4b/H-6 assigned the C-1 and C-6 relative configurations as drawn.

Compound **2** was given a molecular formula same as **1** based on the HR-ESIMS ion at *m*/*z* 385.1261 ([M + Na]^+^, calcd for 385.1258), being an isomer of the latter. The NMR data (Appendix A) for **2** were highly similar to those of **1**, and the only difference was attributed to the presence of a (*Z*)-coumaroyloxy group in the former, replacing the (*E*)-counterpart in the latter, which was evidenced by the resonances at *δ*_C_ 167.8, 160.2, 145.5, 133.7 (2C), 127.6, 116.1, 115.9 (2C), together with *δ*_H_ 7.64, 6.76 (both d and 2H, *J* = 8.7 Hz), 6.90, 5.82 (both d, *J* = 12.8 Hz). Subsequent examination of 2D NMR spectra (see Appendix A) confirmed the *E*/*Z* isomeric relationship and the identical relative stereochemistry for the two co-metabolites.

Compound **3** had the molecular formula C_18_H_20_O_7_ as determined by NMR data and HR-ESIMS analysis at *m*/*z* 371.1100 ([M + Na]^+^, calcd for 371.1101), being 14 mass units less than that of **1** and indicative of a demethyl analogue of the latter. Analysis of the NMR data (Appendix A) for **3** corroborated this hypothesis, as evidenced by the disappearance of the diagnostic 1-OMe signals in **3** compared with **1**. Moreover, C-1 in **3** was significantly up-field shifted by 7.5 ppm, whereas H-1 in **3** was down-field shifted by 0.42 ppm. The structure assignment for **3** as the 1-*O*-demethyl derivative of **1** was also verified by the interpretation of full 2D NMR data (see Appendix A).

A molecular formula same as **3** was assigned to compound **4** by HR-ESIMS analysis at *m*/*z* 371.1095 ([M + Na]^+^, calcd for 371.1101). The NMR data (Appendix A) for **4** were very close to those of **3**, and as with the case of **1** and **2**, compound **4** was also identified to be the *Z*-isomer of **3**. This structure assignment for **4** was further confirmed by examination of full 2D NMR data (see Appendix A).

Compound **5** was given the molecular formula C_20_H_26_O_8_ based on NMR data and the HR-ESIMS ion at *m*/*z* 417.1523 ([M + Na]^+^, calcd for 417.1520). In addition to the (*E*)-coumaroyloxy unit as in **1**, the NMR data (Appendix A) for **5** also showed the presence of two acetal methines [*δ*_C_ 99.7, 99.1; *δ*_H_ 5.01 (d, *J* = 5.5 Hz), 4.77 (dd, *J* = 8.6, 4.6 Hz)], an oxygenated sp^3^ quaternary carbon (*δ*_C_ 80.0), an sp^3^ oxymethine [*δ*_C_ 77.3; *δ*_H_ 4.14 (ddd, *J* = 6.8, 6.1, 4.6 Hz)], an sp^3^ oxymethylene [*δ*_C_ 71.1; *δ*_H_ 4.22, 4.09 (both d, *J* = 11.1 Hz)], two methoxy groups [*δ*_C_ 56.1, 55.8; *δ*_H_ 3.464 (s) 3.462, (s)], and a series of non-oxygenated aliphatic units (*δ*_C_ 48.3, 46.7, 44.3, 32.8; *δ*_H_ 2.22, 2.18 (2H), 2.04, 1.83, 1.56]. These NMR features were consistent with those of **1**, except for the appearance of an extra methoxy unit (3-OMe). Further inspection of ^1^H–^1^H COSY data (Figure 2) revealed a structural fragment of –CHCHCH(CHCH_2_)CH_2_CH– (H-1/H-9/H-5/(H-6/H_2_-7)/H_2_-4/H-3), which was subsequently linked to other structural moieties and/or quaternary carbon(s) by the HMBC signals from 1-OC*H*_3_ and H-3 to C-1, 3-OC*H*_3_ to C-3, and H_2_-10 to C-7, C-8, C-9, and C-9′. The constitution structure of **5** was thus constructed, and the difference of **5** from **1** was that the epoxy bond linking C-3 and C-8 in the latter was cleaved and replaced by the 8-OH and 3-OMe groups in the former. The relative configuration of **5** was established to be identical with that of **1** by interpretation of NOESY and 1D-NOE data (Figure 2). Due to the severe overlapping of some key signals in the ^1^H NMR spectrum in methanol-*d*_4_, compound **5** was then hydrolyzed with NaOH to acquire a coumaroyl-removing product **5r**, whose NOESY and 1D-NOE spectra in CDCl_3_ were obtained (see Appendix A). The most crucial NOESY correlation of H-4b/H-7a supported the *cis*-conjunction for the bicyclic ring system of **5r**, and then the two protons were assigned to be α-orientated, with H-5 and H-9 being β-directed. Other key NOESY and NOE signals included those of H-4b/H-1, H-4b/H-6, H-6/H-7a, H-7b/H-10, H-10/H-9, H-3/H-4a, and H-4a/H-5, indicating the relative configurations at C-1, C-3, C-6, and C-8 as shown.

Compound **6** was assigned a molecular formula same as **5** by HR-ESIMS analysis at *m*/*z* 417.1525 ([M + Na]^+^, calcd for 417.1520). Careful comparison of the NMR data (Appendix A) for **6** with those for **5** revealed high similarity between the two co-isolates, and the only difference was attributed to the replacement of the (*E*)-coumaroyloxy group in the latter by a (*Z*)-coumaroyloxy group in the former. As with the case of **1** and **2**, compound **6** was identified to be the *Z*-isomer of **5**, which was further confirmed by examination of full 2D NMR data (see Appendix A).

Compound **7** was assigned the molecular formula C_20_H_24_O_7_ by NMR data and HR-ESIMS analysis at *m*/*z* 399.1414 ([M + Na]^+^, calcd for 399.1414), being 18 mass units less that of **5** and indicative of a dehydration analogue. Detailed inspection of the NMR data (Appendix A) for **7** authenticated this inference, with diagnostic signals for the –CH(OH)CH_2_– (C-6/C-7) moiety in **5** being replaced by those for a 1,2-disubstituted double bond [*δ*_C_ 138.9, 134.8; *δ*_H_ 6.00 (dd, *J* = 5.7, 2.6 Hz), 5.81, dd (*J* = 5.7, 1.5 Hz)] in **7**, which was also supported by analyses of ^1^H–^1^H COSY and HMBC data (see Appendix A). The relative configurations for all the chiral carbons in **7** were determined to be consistent with those for the respective counterparts in **5** based on a careful examination of ^1^H–^1^H couplings and NOESY data (see Appendix A).

Compound **8** had a molecular formula same as **5** as determined by HR-ESIMS analysis at *m*/*z* 417.1517 ([M + H]^+^, calcd for 417.1520), being isomeric with the latter. The NMR data (Appendix A) for **8** were close to those for **5**, revealing common structural features between the two co-isolates. Further analyses of ^1^H–^1^H COSY and HMBC data (see Appendix A) demonstrated that the two compounds possessed identical planar structures and were a pair of stereoisomers. The relative configuration of **8** was assigned as shown by the interpretation of ^1^H–^1^H couplings and NOESY data (see Appendix A). The key cross-peak of H-3/H-6 supported their co-facial relationship and the *cis*-conjunction of the bicyclic core, and then the correlations of H-6/H-7a, H-7b/H-5, H-7b/H_2_-10, H_2_-10/H-9, as well as the close *J*_1,9_ value of its hydrolysis product **8r** (5.0 Hz) to that of **5r** (4.9 Hz) in methanol-*d*_4_, established the remaining chiral centers as shown. Compound **8** was thus characterized as the 3-epimer of **5**.

It is apparent that compounds **1**–**6** and **8** all had only one UV chromophore, which is not vicinal to any of the chiral centers, and thus they did not show discernible Cotton effects in the ECD measurements. Therefore, it is not possible to assign their absolute configurations via ECD-related methods [16]. As compounds **5** and **8** were obtained in enough amounts, they were chosen for further chemical derivatization. The decoumaroyl products **5r** and **8r** were first obtained via alkaline hydrolysis of **5** and **8**, respectively, and then the (*S*) and (*R*)-MTPA esters of them were prepared. By applying the Mosher’s NMR method [17], the C-6 configurations in both **5r** and **8r** were thus established to be *R* (Appendix A). In addition, the absolute configurations at C-8 in the two molecules were both determined to be *R* based on the negative Cotton effects at 312 and 311 nm (Appendix A), respectively, in the in situ Mo_2_(OAc)_4_ induced ECD experiments [18]. The two different methods confirmed the absolute stereochemistries of **5** and **8** as shown, and based on a biogenetic consideration, the absolute configurations of other iridoid co-metabolites were also assigned.

Compound **9** was given the molecular formula C_21_H_28_O_8_ based on NMR data and the HR-ESIMS ion at *m*/*z* 443.1481 ([M + Cl]^–^, calcd for 443.1478). Analysis of the NMR data (see Section 2.3.9) for **9** revealed characteristic signals of a 1,3,4-trisubstituted benzene unit [*δ*_C_ 151.7, 147.2, 137.9, 121.8, 119.3, 113.9; *δ*_H_ 6.81 (d, *J* = 8.2 Hz), 6.78 (d, *J* = 1.9 Hz), 6.66 (dd, *J* = 8.2, 1.9 Hz)], a 1,3,4,5-tetrasubstituted and symmetrical benzene unit [*δ*_C_ 148.9 (2C), 135.9, 133.3, 105.6 (2C); *δ*_H_ 6.69 (2H, s)], three methoxy groups [*δ*_C_ 56.7 (2C), 56.4; *δ*_H_ 3.80 (6H, s), 3.77 (s)], two sp^3^ oxymethines [*δ*_C_ 86.4, 74.3; *δ*_H_ 4.80 (d, *J* = 6.0 Hz), 4.30 (m)], two sp^3^ oxymethylenes [*δ*_C_ 62.3, 62.2; *δ*_H_ 3.85 (dd, *J* = 11.9, 5.8 Hz), 3.77 (m), 3.54 (2H, t, *J* = 6.5 Hz)], and two normal aliphatic methylenes [*δ*_C_ 35.5, 32.7; *δ*_H_ 2.59 (2H, *J* = 7.7 Hz), 1.79 (2H, tt, *J* = 7.7, 6.5 Hz]. These NMR features were very similar to those of (7*R*,8*R*)-*threo*-4,7,9,9′-tetrahydroxy-3,5,3′-trimethoxy-8,4′-oxyneolignane isolated from *Alangium chinense* [19], suggesting a neolignan analogue. Subsequent inspection of 2D NMR ^1^H–^1^H COSY and HMBC data (Figure 2) constructed a constitution structure for **9**, identical to that for the aforementioned known neolignan. By comparing the NMR resonances for the two molecules, the main differences were observed for signals of the C-7 to C-9 fragment, implying that compound **9** was a stereoisomer of *erythro* relative configuration. This structure assignment was further confirmed by analyzing the absolute Δ*δ* values for H-9a/H-9b (0.06 ppm) [20] and C-7/C-8 (12.1 ppm) [21] of **9**. In an attempt to establish the absolute stereochemistry of **9**, the nearly smoothed ECD curve with no obvious Cotton effects alerted us of the possible presence of racemate, according to our previous experiences [22]. The subsequent chiral HPLC analysis and separation indeed afforded a pair of enantiomers (**9a** and **9b**) with antipodal optical rotations and Cotton effects. Finally, the 8*S* and 8*R* absolute configurations for **9a** and **9b** were established by their (+) and (–) Cotton effects, respectively, at 243 nm in the ECD spectra (Figure 3) [19,23].

### 3.2. Structure Characterization of Known Compounds

Besides the new compounds described above, thirteen known isolates were also obtained in the present work. These compounds, on the basis of NMR analyses and the comparison with those of known structures in the literature, were identified as *erythro*-4,7,9,9′-tetrahydroxy-3,3′-dimethoxy-8,4′-oxyneolignane (**10**) [24], *threo*-4,7,9,9′-tetrahydroxy-3,3′-dimethoxy-8,4′-oxyneolignane (**11**) [24], cedrusin (**12**) [25,26], 4-*O*-methylcedrusin (**13**) [27], dihydrodehydrodiconiferyl alcohol (or 3′-*O*-methylcedrusin, **14**) [28], 5-methoxydihydrodehydrodiconiferyl alcohol (**15**) [29], polystachyol (**16**) [30], neo-olivil (**17**) [31], icariol A_2_ (**18**) [32,33], megastigmane (**19**) [34], bursephenylpropane (**20**) [35], tetrahydrodehydrodiconiferyl alcohol (**21**) [36,37], and 3-deoxyartselaenin C (**22**) [38]. All these constituents from the title plant mainly belong to two structural classes, and details of their structural types are listed in Appendix A.

During the structure characterization of the known chiral compounds, the [α]_D_ and ECD data of some lignans showed obvious deviations from those reported formerly. According to a very recent review on natural enantiomers [22], lignans are among the structural types that very likely exist in racemate or scalemic mixtures, and also inspired by the example of **9**, the remaining known lignans were further subjected to chiral HPLC analysis. As expected, compounds **10–18** were at least partially racemic, and subsequent separation tried on different chiral columns successfully afforded pure enantiomers of all of them. Lignans are widely distributed in nature and have been isolated from plants of 70+ families and thus represent one of the most-encountered classes of natural products [39]. However, owing to the structural diversity and similarity, the structures of many lignans, especially those reported in early years, were often wrongly assigned. For instance, compound **14** was erroneously characterized as **13** in some reports [40,41]. Another issue in the chemistry of lignans is that most researchers used to only focus on the characterization of constitution structure and relative configuration but tended to neglect the optical purity and absolute stereochemistry, and this bias is particularly true for those seemingly ‘known’ compounds, because we usually prefer to see ‘new chemistry’ (new structure). There were thus some questions yet to be answered in the previous isolation and structure identification of the aforementioned known lignans. For example, polystachyol (**16**) was obtained apparently as a racemate in 2006 [30] and a partial racemate in 2012 [42] based on the [α]_D_ values of ~0 and +3.3, respectively, but no further chiral analysis or separation was carried out. In addition, an enantiomer of **12** was described in 2005 by Kim and co-workers [26], whereas only selective Cotton effects in the ECD spectrum were reported. Herein, we reported the full ECD data (for Cotton effects see Appendix A and for ECD spectra see Figure 3) of these lignan enantiomers for the convenience of future researchers to have a better comparison, as well as a brief description of their absolute configuration assignments (how they were differentiated from each other).

As with the case of **9a**/**9b**, the 8*R* and 8*S* absolute configurations for **10a** and **10b** were determined by the (–) and (+) Cotton effects at 239 nm, respectively [19,23]. Similarly, compounds **11a** and **11b** were assigned 8*R* and 8*S* absolute configurations based on their respective (–) and (+) Cotton effects at 236 nm. For the benzofuran-type lignans, molecule **12a** was determined to have the 7*S*,8*R*-configuration on the basis of its (+) and (–) Cotton effects at 293 (^1^L_b_ band) and 227 (^1^L_a_ band) nm, respectively [24,43], and its enantiomer **12b** thus had the 7*R*,8*S*-configuration. Compounds **13a**/**13b** and **14a**/**14b** showed highly consistent ECD curves with the corresponding enantiomers **12a**/**12b**, and their absolute configurations were hence established as shown. Owing to the introduction of an additional methoxy group, the ECD spectra of **15a**/**15b** exhibited some variations compared to those of **12**–**14**, and their absolute configurations were assigned as drawn via the Cotton effects at 292 nm [26,43]. Compounds **16a** and **16b** were differentiated from each other and assigned the drawn absolute configurations by comparing their ECD curves with the computed ones (Figure 3) obtained from a time-dependent density functional theory (TD-DFT) method. The absolute configuration of **17b** was originally established as 7*R*,8*S*,7′*R*,8′*S* via chemical degradation to compounds of known chirality [31]; thus, its enantiomer **17a** had the 7*S*,8*R*,7′*S*,8′*R*-configuration. Moreover, compound **18a** was originally assigned the drawn absolute configuration by the ECD exciton chirality method [44], and its enantiomer **18b** was then determined to have the opposite stereochemistry. Interestingly, **18** was first reported in the (–)-enantiomeric form with the trivial name icariol A_2_ [32] but was later isolated as a racemate and given another name, huazhongilexin [45]. It is also interesting to note that compound **21** was separated, without acquisition of its enantiomer, from the mixture of **12a**/**12b** via chiral HPLC separation, and only one previous report described the separation of both enantiomers but without determining the absolute configuration [36]. In the present work, we assigned the *R*-configuration to **21** based on the TD-DFT-based ECD calculation (Figure 3). Moreover, as the NMR data for **21** in methanol-*d*_4_ in the literature [37] were assigned via analyses of 2D NMR data without support from direct 1D NMR data, we then report them again in this study (See Section 2.3.10).

### 3.3. Biological Evaluations

#### 3.3.1. Antioxidant Evaluation

The antioxidant activities of extracts acquired through different methods from *V. bracteatum* have often been assessed in previous reports [2], and several flavonoids were reported to be responsible for the effect [46]. However, in another study by Su et al., the antioxidant capacity of the VBL extract was observed to have no direct relationship with the flavonoid content and could also be influenced by other constituents [47]. We thus tested the antioxidant effects of all the isolates obtained in the present work by assessing their free radical scavenging ability with ABTS and DPPH assays. As shown in Table 1, all the screened compounds except **19** and **20** showed mild to significant capturing activity against ABTS radicals with IC_50_ values ranging from 4.36 to 168.0 μM. While more than two thirds of the molecules exerted better activity than the positive control ascorbic acid (IC_50_ = 24.29 ± 0.41 μM), compounds **9a**, **12a**, **17a**/**17b**, **18a**/**18b**, and **21** performed particularly well, with IC_50_ values below 10 μM. Meanwhile, all the iridoid constituents (**1**–**8**) did not exhibit a scavenging effect toward DPPH radicals, whereas most lignans showed moderate activity.

#### 3.3.2. Neuroprotective Evaluation

The MTT assay was used to determine the viability of PC12 neuron cells to evaluate the neuroprotective activity of the eight new iridoids **1–8**, and the promotion or inhibition on cell growth of these compounds was first excluded in the testing concentration range by the same assay. The 6-OHDA-induced cell injury was first evaluated in the range of 50–250 μM, and 150 μM with an inhibition ratio of 52.68% was chosen for the final experiment (Figure 4A). The neuroprotective effects of **1–8** toward the PC12 cell injury were then evaluated, and as shown in Figure 4B, iridoids **1**, **5**, **6**, and **7** exerted apparent protective activity at 50 μM, with **1** being comparable to the positive control NAC.

#### 3.3.3. α-Glucosidase Inhibitory Evaluation

The hypoglycemic effect, an important strategy to control diabetes, is the most well-known and investigated health benefit of *V. bracteatum*, and this biological property of VBL extract and its polysaccharide constituents was confirmed in diabetic mice by reducing hepatic gluconeogenesis to enhance glucose metabolism [2]. In the current study, the inhibitory assay toward an important diabetic target α-glucosidase was also carried out on all the isolates (Table 2). Compound **11a** showed the best inhibitory effect with an IC_50_ of 27.75 ± 2.82 μM, and compounds **10b** (IC_50_ = 39.96 ± 9.71 μM), **11b** (IC_50_ = 61.80 ± 3.58 μM), **12a** (IC_50_ = 69.37 ± 3.32 μM), and **18a** (IC_50_ = 46.36 ± 7.92 μM) also exhibited significant inhibition against the enzyme, while compounds **9a**/**9b**, **10a**, **13a**, **16a**/**16b**, **17a**/**17b**, and **18b** were less active, with IC_50_ values ranging from 106.3 to 311.9 μM.

#### 3.3.4. Anti-Inflammatory Evaluation

The anti-inflammatory properties of different extracts from *V. bracteatum*, by evaluating their inhibition against cyclooxygenase-2 and NO production, are also documented in the literature [2]. We thus screened all these isolates in the NO production inhibitory assay with a lipopolysaccharide-induced model in murine RWA264.7 macrophages, and only compound **14a** displayed mild activity (IC_50_ = 52.16 ± 6.17 μM).

#### 3.3.5. Antimicrobial and AChE Inhibitory Evaluations

The preservative effect of VBL extract on food has long been known and applied by local residents [2], and this might be related to the antimicrobial activity of some chemical constituents in the plant. The water extract of VBL was recently assessed for antibacterial activity against *S. aureus*, *B. subtilis*, and *E. coli* with positive feedback [48]. In addition to the three aforementioned strains, *P. aeruginosa* was further included in the present study’s antibacterial assays on all the molecules, but none of them displayed decent activity. Moreover, the inhibition of all these compounds against AChE, an important neurodegenerative disease target, was also screened, whereas they were all inactive.

## 4. Discussion

Iridoids represent a unique group of natural products of the monoterpene family, and they are mainly found in dicotyledonous plants [49]. Former phytochemical investigations on the iridoid ingredients from *V. bracteatum* afforded only several glycosides, and no free aglycones were reported [2]. Eight free iridoids were obtained in the present study, and compounds **1**–**4** feature a novel revolving-door shaped scaffold. The reason of this constituent difference could be attributed to the extracting solvent and the research focus, as more polar solvent systems (mainly MeOH-H_2_O) were used in the most previous reports, and the separation work generally concentrated on the polar compounds. As for the lignan metabolites from *V. bracteatum*, there were only three members counted in the 2021 review [2], and we did not find other reports in a further literature retrieval. Thus, this is the first time that such a big number of lignans was recorded from this plant. Moreover, the wide natural occurrence of lignan enantiomers was discussed in the structure characterization section, which was often neglected in early documents and has only been concerned in recent years [22]. In this study, we checked the enantiomeric purities of all lignans and then successfully separated the enantiomers by different chiral columns. Furthermore, we unambiguously established the absolute configurations of all these lignan molecules based on ECD analyses and clarified some improper or erroneous structural assignments in previous reports.

Oxidative stress, a resultant physiological state from the imbalance of the redox system, is usually accompanied by the accumulation of destructive free radicals (or reactive species) and decreasing protection from antioxidant defense [50], and it is generally considered to play an important role in the development of many chronic diseases such as diabetes, cancers, and neurological (i.e., dementia) and cardiovascular (i.e., atherosclerosis) disorders, many of which are age-related [2,51]. Therefore, tremendous efforts have been put into the study of oxidative prevention and related illnesses, and increasing evidence has indicated that antioxidants could reduce the damage caused by oxidative stress by quenching the activity or inhibiting the formation of free radicals, thus enhancing immunity and increasing healthy longevity [51,52]. In addition, the efficacy of antioxidants to slow down the progressive deterioration in neurodegenerative diseases has also been widely explored [53,54]. In a very recent review, the role of plant-derived natural antioxidants by reducing oxidative stress to exert anti-inflammatory, anti-aging, and neuroprotective properties has been well discussed [55]. In the present project, thirty-two small-molecule chemical constituents, including ten new ones, were isolated and structurally elaborated from VBL, which not only greatly enriches the metabolite diversity of the title plant but also represents a significant progress in its chemical study. More excitingly, nearly all and about half of the isolates showed decent antioxidant activity in the ABTS and DPPH radical scavenging assays, respectively. In addition, selective new iridoids also exhibited noticeable neuroprotection against the 6-OHDA-induced PC12 cell injury. These results demonstrate that *V. bracteatum* is a rich source of natural antioxidants, and lignans and iridoids, in addition to the previously described flavonoids, could also make an important contribution to its antioxidation-relevant health benefits.

## 5. Conclusions

In summary, both crude extracts and specific chemical constituents from VBL exhibit a variety of bioactivities related to its traditional health-beneficial applications [2]. Although it is well known that the production and content of the bioactive metabolites in a plant vary in pace with different habitats and growth stages, all former studies have at least partially demonstrated the non-toxic and health-promoting properties of the title species, which provides a fundamental and solid evidence to develop new food products with health benefits from VBL.

In the present study, we obtained 32 small molecule constituents including ten new ones (**1**–**9**) and ten pairs of enantiomers (**9**–**18**) from VBL, and their chemical structures and absolute configurations were determined by NMR, HR-ESIMS, and ECD analyses, as well as chemical derivatization and degradation. By screening these compounds in a panel of bioassays, the antioxidant activities of most isolates, along with the neuroprotective, α-glucosidase inhibitory, and NO release inhibitory effects of selective molecules, were validated. The current work expanded the knowledge on both the chemistry and bioactivity of the constituents from *V. bracteatum* and further consolidated our understanding toward this plant as a source of functional food. All in all, there is still a lot more to be explored and exploited regarding this fascinating herbal species.

## Figures and Tables

**Figure 1 foods-12-00177-f001:**
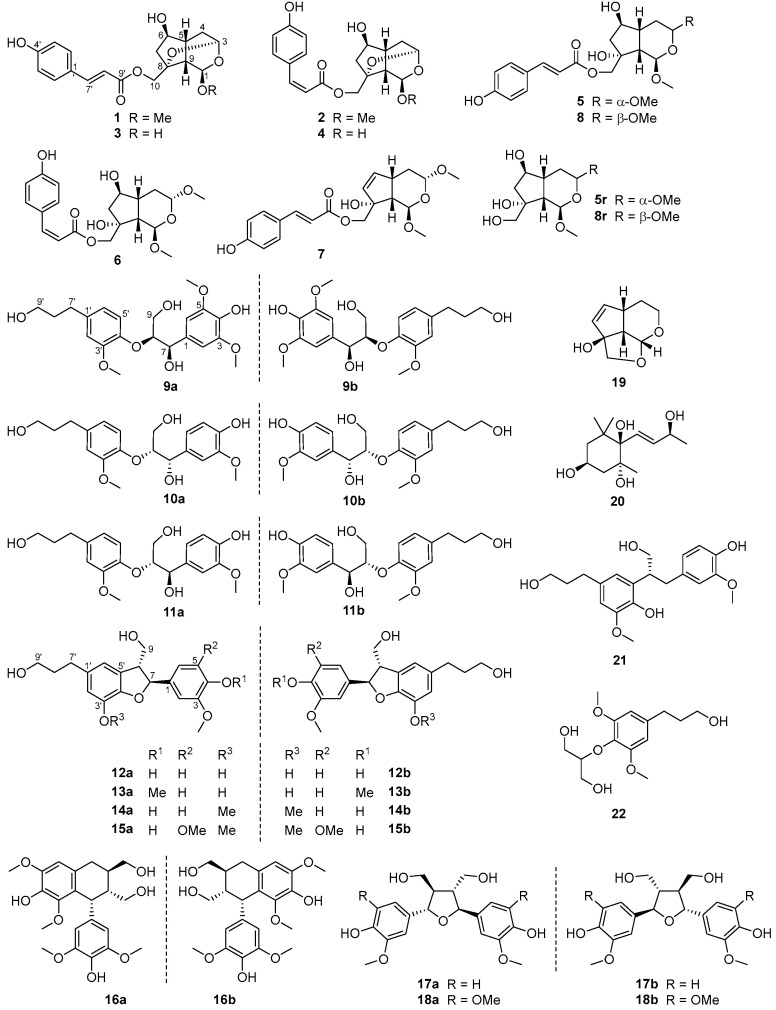
Structures of the chemical constituents from *V. bracteatum*.

**Figure 2 foods-12-00177-f002:**
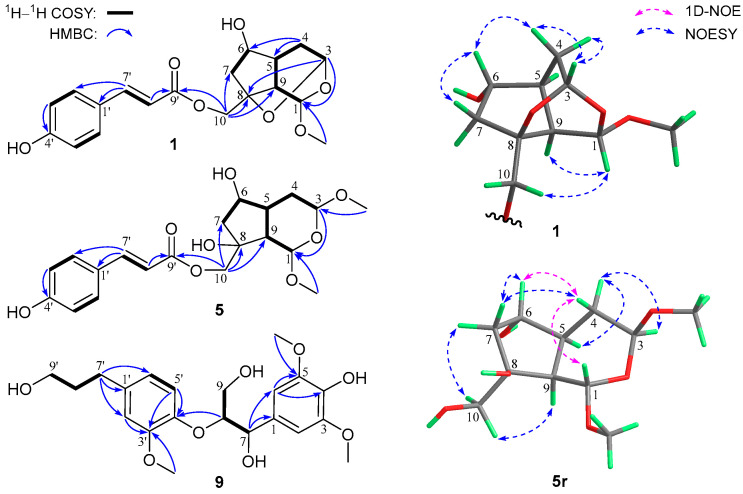
Key 2D NMR correlations for **1**, **5**, **5r**, and **9** (for a better illustration, the enantiomers of **1** and **5r** were used to show the NOE/NOESY signals).

**Figure 3 foods-12-00177-f003:**
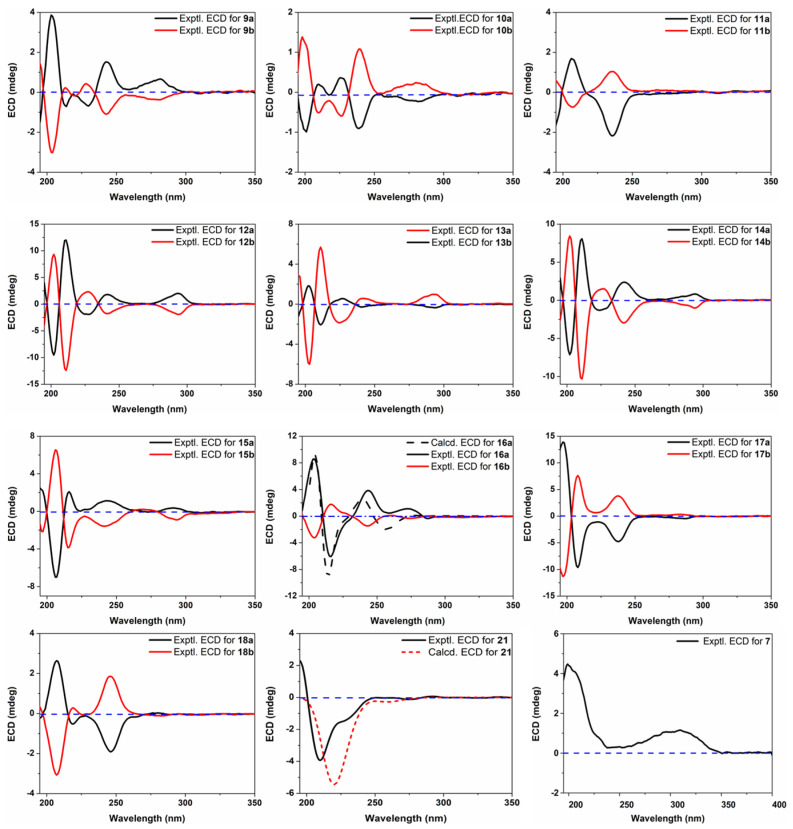
Experimental and calculated ECD spectra for **7**, **9**–**18**, and **21**.

**Figure 4 foods-12-00177-f004:**
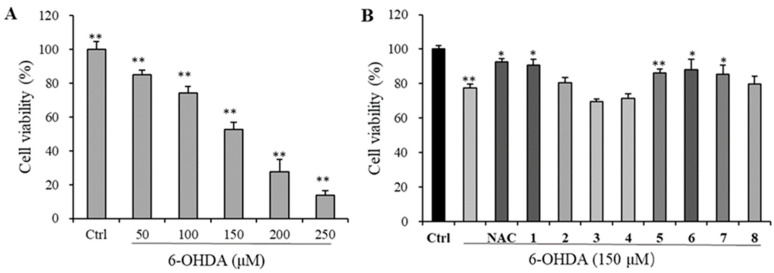
Neuroprotective evaluation of iridoids **1**–**8** in PC12 cells. (**A**) The cell viability of PC12 cells upon 6-OHDA treatment in the concentration range of 50–250 μM; (**B**) neuroprotective effects of **1**–**8** at 50 μM against 6-OHDA-induced PC12 cell injury. Data are expressed as mean ± SD from triplicate experiments. * *p* < 0.05, ** *p* < 0.01, compared with the control group.

**Table 1 foods-12-00177-t001:** The free radical scavenging activity of the constituents from *V. bracteatum*
^a^.

Compounds	ABTS	DPPH	Compounds	ABTS	DPPH	Compounds	ABTS	DPPH
**1**	15.28 ± 1.37	NA	**9a**	9.39 ± 0.97	47.49 ± 7.24	**14a**	12.32 ± 1.82	NA
**2**	34.17 ± 3.03	NA	**9b**	10.08 ± 0.67	52.62 ±6.89	**14b**	14.07 ± 1.30	NA
**3**	35.58 ± 1.04	NA	**10a**	19.69 ± 0.61	NA	**15a**	17.21 ± 0.89	40.43 ± 6.09
**4**	19.40 ± 0.24	NA	**10b**	16.23 ± 2.65	NA	**15b**	24.25 ±3.92	55.10 ± 2.43
**5**	32.01 ± 1.76	NA	**11a**	12.13 ± 1.42	98.65 ± 1.78	**16a**	11.09 ± 3.31	47.92 ± 6.60
**6**	28.25 ± 0.63	NA	**11b**	14.79 ± 1.25	105.3 ± 4.11	**16b**	26.20 ± 2.24	96.16 ± 8.34
**7**	50.65 ± 2.97	NA	**12a**	9.54 ± 0.64	31.15 ± 1.90	**17a**	5.96 ± 0.30	40.79 ± 5.98
**8**	39.14 ± 2.30	NA	**12b**	12.30 ± 0.24	44.28 ± 2.05	**17b**	7.75 ± 0.71	62.85 ± 5.65
**19**	NA	NA	**13a**	21.66 ± 3.50	NA	**18a**	6.87 ± 0.54	17.32 ± 7.51
**20**	NA	NA	**13b**	37.05 ± 8.17	NA	**18b**	4.36 ± 0.30	17.92 ± 4.58
**21**	9.17 ± 0.24	NA	**22**	168.0 ± 4.34	20.21 ± 3.53	Ascorbic acid	24.29 ± 0.41	13.30 ± 1.86

^a^ Data are presented as IC_50_ in μM in the form of mean ± SD (n = 3). ‘NA’ means ‘not active’ (<50% radical scavenging activity at the highest testing concentration of 200 μM).

**Table 2 foods-12-00177-t002:** α-Glucosidase inhibitory assay results for all the compounds ^a^.

Compounds	IC_50_ (μM)	Compounds.	IC_50_ (μM)	Compounds	IC_50_ (μM)
**1**	NA	**10b**	39.96 ± 9.71	**16a**	121.1 ± 3.18
**2**	NA	**11a**	27.75 ± 2.82	**16b**	243.4 ± 4.24
**3**	NA	**11b**	61.80 ± 3.58	**17a**	284.6 ± 4.53
**4**	NA	**12a**	69.37 ± 3.32	**17b**	106.31 ± 12.86
**5**	NA	**12b**	NA	**18a**	46.36 ± 7.92
**6**	NA	**13a**	293.9 ± 26.87	**18b**	212.9 ± 15.98
**7**	NA	**13b**	NA	**19**	NA
**8**	NA	**14a**	NA	**20**	NA
**9a**	311.9 ± 10.18	**14b**	NA	**21**	NA
**9b**	304.6 ± 6.64	**15a**	NA	**22**	NA
**10a**	214.4 ± 16.33	**15b**	NA	Acarbose	493.5 ± 8.62

^a^ Data are presented as IC_50_ in μM in the form of mean ± SD (n = 3). ‘NA’ means ‘not active’ (<50% inhibition at the highest testing concentration of 500 μM).

## Data Availability

The data are available on request from the corresponding author.

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
