# Peer review of "Identification of Small-Molecule Bioactive Constituents from the Leaves of Vaccinium bracteatum Confirms It as a Potential Functional Food with Health Benefits"

_foods, 2023, doi:10.3390/foods12010177_

Round 1
Reviewer 1 Report
Dear Authors,
In general I highly asses Your article. However as a scientific referee I have some comments to it:
Page 1: Please explain the abbreviations in the Abstract - NMR and ECD, the same for DPPH and ABTS. Since the Abstract should be read independently from the whole manuscript it is better to explain them.
Page 10: In my opinion the paragraph in Discussion (lines 360 - 382) can be moved into the Introduction. However, it is only my suggestion, so perhaps the Authors should decide about this.
Page 11: In Extraction and isolation - please provide information about purity class of water used for the extractions of plant material (line 413). The type of water applied in the experiments may seriously affect the results obtained during the studies.
Additional issues:
1. The manuscript is scientific.
2. The method part of the manuscript is appropriate.
3. In my opinion the main question in the research was revealing the role of small molecule constituents in functional food properties of Vaccinium bracteatum species. It is relevant and interesting.
4. In my opinion the topic can be original for people interested in natural medicines.
Compared with other published material, The paper adds a new knowledge on small-molecule bioactive compounds present in the leaves of Vaccinium bracteatum and about analytical techniques used to study them.
5. The paper is well written, but as I wrote above: In my opinion the paragraph in Discussion (lines 360 - 382) can be moved into the Introduction.
6.The text is clear and easy to read in general.
The conclusions are consistent with the evidence and arguments presented.
They addressed the main question posed.
General evaluation: minor revision
Author Response
Response to Reviewer #1
- Comment: Page 1: Please explain the abbreviations in the Abstract - NMR and ECD, the same for DPPH and ABTS. Since the Abstract should be read independently from the whole manuscript it is better to explain them.
Response: Thanks for the constructive suggestion. We have added the full words of the abbreviations in the Abstract.
- Comment: Page 10: In my opinion the paragraph in Discussion (lines 360 - 382) can be moved into the Introduction. However, it is only my suggestion, so perhaps the Authors should decide about this.
Response: Thanks for the suggestion. After careful consideration of the content of the whole manuscript, we have decided not to move this paragraph to the Introduction based on the following reasons. The Discussion part usually only presents and re-emphasize some key and meaningful points of the whole work, while the Introduction part is a highly condensed summary of all the content of the work. Since antioxidant activity is only part of the biological properties of the acquired constituents, it is thus more appropriate to leave this paragraph in the Discussion section rather than move it to the Introduction section.
- Comment: Page 11: In Extraction and isolation - please provide information about purity class of water used for the extractions of plant material (line 413). The type of water applied in the experiments may seriously affect the results obtained during the studies.
Response: Thanks for the reviewer’s carefulness. We have used distilled water for the partition process, and “distilled” has been added as suggested.
- Lastly, we are greatly encouraged by the positive comments from the reviewer and appreciate his efforts very much in improving the quality of our manuscript.
Reviewer 2 Report
I would like to thank the authors for the outstanding effort they put into the article.
Identification of small-molecule bioactive constituents from the leaves of Vaccinium bracteatum confirms it as a potential functional food with health benefits
Is it possible to put the compounds in a table and indicate to which group they belong?
I want to indicate in front of each group of separate compounds which of these groups follow flavones, glycosides, anthocyanins, terpenoids, iridescent glycosides, phenylpropanoids and lignans. So that the reader is aware of the nature of the compounds separated in the descriptive part of the compounds.
Page 6 Structure characterization of known compounds. Line212
Author Response
Response to Reviewer #2
- Comment: Is it possible to put the compounds in a table and indicate to which group they belong?
I want to indicate in front of each group of separate compounds which of these groups follow flavones, glycosides, anthocyanins, terpenoids, iridescent glycosides, phenylpropanoids and lignans. So that the reader is aware of the nature of the compounds separated in the descriptive part of the compound.
Structure characterization of known compounds. Line21 Page 6
Response: Thanks for the constructive suggestion. We have put all the compounds and their structural classes in Table S4 of the Supplementary materials and added one more descriptive sentence to the text. (Section 2.2 in page 6).